# Trends in Asymptomatic Nasopharyngeal *Streptococcus pneumoniae* Carriage with qPCR and Culture Analysis

**DOI:** 10.3390/microorganisms10102074

**Published:** 2022-10-20

**Authors:** Julie-Anne Lemay, Leah J. Ricketson, James D. Kellner

**Affiliations:** 1Department of Pediatrics, University of Calgary, Calgary, AB T2N 1N4, Canada; 2Department of Community Health Sciences and Department of Microbiology, Immunology & Infectious Diseases, University of Calgary, Calgary, AB T2N 1N4, Canada; 3Alberta Children’s Hospital Research Institute, Calgary, AB T3B 6A8, Canada

**Keywords:** *Streptococcus pneumoniae*, 13-valent pneumococcal conjugate vaccine, nasopharyngeal carriage

## Abstract

We previously reported trends in pneumococcal nasopharyngeal carriage in the post-PCV13 era as detected by conventional culture methods. Our current aim is to assess if there are fundamental differences in the clinical and demographic features of children who have pneumococcal carriage detected by qPCR compared with culture analysis. The CASPER team conducted point-prevalence surveys in 2016 in healthy children in Calgary to determine trends in overall and serotype-specific pneumococcal nasopharyngeal carriage. Being 18 months of age (*p* = 0.009), having at least one sibling under 2 years of age (*p* = 0.04), having only sibling(s) over 2 years of age (*p* = 0.001), and childcare attendance (*p* = 0.005) were associated with carriage by qPCR methods only. Having only sibling(s) older than 2 years of age was associated with carriage detected by both qPCR and culture methods (*p* = 0.001). No clinical factors were associated with carriage detected by both qPCR and culture compared to qPCR methods only. Both analyses are suitable methods to detect carriage; however, qPCR analysis is more sensitive and more cost-effective. As there are no fundamental differences in the children that have pneumococcal nasopharyngeal carriage detectable by qPCR methods compared to conventional culture methods, molecular analysis may be a preferable option for future carriage studies.

## 1. Introduction

Asymptomatic carriage of *Streptococcus pneumoniae* (pneumococcus) is common in healthy children, with carriage preceding pneumococcal disease [1]. It can cause invasive or non-invasive infections, and young children continue to be an important reservoir for circulation of *S. pneumoniae* in the community [2].

In Alberta, the seven-valent pneumococcal protein–polysaccharide conjugate (PCV7) vaccine was introduced for routine use in infants in 2002 with three doses given before one year of age and a single booster dose after one year (3 + 1 schedule) [3]. PCV7 was replaced by the 13-valent pneumococcal protein–polysaccharide conjugate vaccine (PCV13) in 2010, with a 2 + 1 schedule in Alberta [3]. Pneumococcal vaccination has led to a decrease in vaccine serotype invasive pneumococcal disease (IPD) and vaccine serotype nasopharyngeal carriage in vaccinated individuals [4,5,6,7]. This has led to a subsequent decrease in vaccine serotype carriage and disease in unvaccinated individuals through herd immunity [8,9,10,11,12,13]. After the introduction of PCV13, the Calgary Area *S. pneumoniae* Epidemiology Research (CASPER) team observed a decrease in both carriage of vaccine serotypes and in overall carriage of *S. pneumoniae* in Calgary, Alberta, as identified by standard culture [4]. However, non-vaccine serotypes were found to have almost completely replaced vaccine serotypes in carriage [4], and the impact on clinical disease still calls for continuous monitoring of *S. pneumoniae* carriage in children.

Laboratory culture to detect S. pneumoniae and Quellung reaction serotyping is the conventional gold standard method to detect *S. pneumoniae* and determine serotypes [14]. However, molecular analysis with various polymerase chain reaction (qPCR) techniques can more frequently and rapidly detect serotype positive samples compared to standard culture methods [15,16] and these methods consistently detect higher levels of pneumococcal carriage compared to culture methods [17,18,19,20,21]. We have previously reported a higher detection rate for *S. pneumoniae* through qPCR methods compared with standard culture, as well as a higher proportion of samples carrying vaccine-serotype strains, demonstrating that vaccine serotypes can still be detected in a considerable proportion of children in the late post-PCV13 era [22].

The purpose of the current study is to assess if there are fundamental differences in the clinical and demographic features of children who have pneumococcal carriage detected by qPCR compared with conventional culture analysis.

## 2. Materials and Methods

### 2.1. Population and Sample Collection

The CASPER team conducted point-prevalence surveys bi-annually during 2003–2005 and annually in 2006, 2010, 2011, 2012, 2014, 2016, and 2018 in healthy children in Calgary to determine overall and serotype-specific trends in pneumococcal nasopharyngeal (NP) carriage. Results from these surveys have been previously published [4,23]. The current study focuses on samples obtained during the 2016 survey (between 11 April 2016 and 2 June 2016). The study population included children visiting Community Health Centers (CHCs) in the Calgary zone of Alberta Health Services (Calgary Health Region prior to 2009) for routine immunizations and who were in one of the following age groups—12 months (±8 weeks), 18 months (±8 weeks), 4.5 years (±6 months). The Calgary zone includes the city of Calgary and surrounding areas, a population of approximately 1,622,391 in 2016 [24]. Routine childhood vaccines, including PCV13, are publicly funded and administered by public health at CHCs in Alberta.

For each survey period, study staff would visit CHCs on days when large numbers of children were scheduled for routine immunizations. A convenience sample of children were approached and invited to participate in the study. Exclusion criteria for recruitment included children with a condition where nasopharyngeal swab collection would be contraindicated (e.g., hemophilia, anatomic anomaly of the upper respiratory tract) and if informed written consent by a parent could not be obtained. Parents of participating children completed a questionnaire to provide demographic and clinical data. A single nasopharyngeal swab was collected from all participating children, using to the World Health Organization described method [25].

### 2.2. Carriage Detection and Serotyping Results

The detailed methods for nasopharyngeal sample collection, culture detection and Quellung serotyping, as well as qPCR detection, using *lytA*, *piaA*, and SP2020 primers (and considered positive if two primers were positive), and qPCR serotyping have been published previously [22], and results have been reutilized for this study. Of note regarding serotyping, isolates that could not be serotyped by Quellung reaction are referred to as non-typeable, whereas strains that did not match one of the thirty-seven serotypes detected by qPCR are referred to as not-typed due to the limitations of the assay. Samples that were culture negative and positive for only one qPCR primer to detect *S. pneumoniae* were considered positive if there was a serotype identified. They were considered equivocal if no serotype was identified. Samples that were both qPCR and culture negative were not serotyped.

### 2.3. Analysis

PCV13 carriage is defined as carriage of PCV13 vaccine and related serotypes (serotypes 1, 3, 4, 5, 6A, 6B, 7F, 9V, 14, 18C, 19A, 19F, 23F, and 6C). For our analysis, the serotypes were further categorized in the following groups: serotypes in the PCV13 vaccine (PCV13), nonvaccine serotypes not included in the PCV13 vaccine (NVT), non-typeable by Quellung for positive samples identified by culture methods (NT), and not-typed by qPCR for positive samples identified by qPCR methods (Not-Typed). For the not-typed samples, none of the 37 serotypes in the qPCR serotyping assay were identified, but it was not known if other defined serotypes were present. However, although the qPCR assay detects only 37 serotypes, these include all PCV13 vaccine serotypes and almost all common and uncommon serotypes causing pneumococcal disease [22]. For further analysis, NT and Not-Typed samples were combined as one group. As previously reported, there were 682 children who participated in the 2016 survey [22], of which 679 were included in our study. Two were excluded as they each received at least one dose of the PCV7 vaccine. One was excluded because of incomplete information recorded. An underlying health condition was defined as the presence of a condition that is considered to put an individual at a high risk of IPD according to the Canadian Immunization Guide [26], as well as any other condition noted by the parent that does not fall under the guide that potentially increases the child’s risk of carriage. Statistical analysis was performed using Stata/SE 17.0 (StataCorp. 2021. Stata Statistical Software: Release 17, College Station, TX: StataCorp LLC). A multinomial logistic regression was used to compare clinical factors between the qPCR and culture positive group (qPCR-pos and Culture-pos) and the qPCR positive only group (qPCR-pos only) to the qPCR and culture negative baseline group (qPCR-neg and Culture-neg). Logistic regression was used to compare clinical factors and serotype groups between the qPCR-pos and Culture-pos group and the qPCR-pos only group. Equivocal results were excluded from all analyses as they could not be classified and represented a minor contribution to overall results. All clinical factors were examined in the regression models and backwards elimination was used to remove factors that were not significant to improve the power of the model for examining the remaining factors. The final model includes factors that are important clinically though some did not show statistical significance.

This study was approved by the Conjoint Health Research Ethics Board (CHREB) at the University of Calgary.

## 3. Results

### 3.1. Identification of S. pneumoniae and Serotype, Demographic and Clinical Factors

As previously reported, there were 682 children who participated in the 2016 survey [22]. In the previous study, from the NP samples collected on all children, 71/682 (10.4%) were culture and qPCR positive, and 243 culture-negative samples were qPCR positive for a total of 314/682 (46.0%) of samples positive for *S. pneumoniae*. In addition, 45/682 samples (6.6%) were equivocal for detection of *S. pneumoniae* [22]. In total, 92/314 (29.3%) of qPCR positive samples had one or more PCV13 serotypes compared to 8.4% culture positive samples that had PCV13 serotypes by Quellung reaction [22]. Table 1 describes the population demographics and clinical features for this survey (of which only 679 participants from the previous study met our inclusion criteria) and the proportions for each factor for qPCR-pos and Culture-pos, qPCR-pos only, and by qPCR-neg and Culture-neg. Most children had no underlying health conditions, no episodes of otitis media in the last year, were not hospitalized overnight in the last 6 months, and were not on antibiotics at the time of the survey (Table 1). The largest age group was the 12-month-old age group. Most children had received two or more doses of the PCV13 vaccine (99%).

### 3.2. Factors Influencing Carriage

Multinomial logistic regression comparing clinical factors associated with qPCR-pos and Culture-pos and qPCR-pos only groups to the qPCR-neg and Culture-neg baseline group is shown in Table 2. Factors significantly associated with having a qPCR-pos only result compared to qPCR-neg and Culture-neg included being 18 months of age versus 12 months of age (*p* = 0.009), having at least one sibling under the age of 2 years old (*p* = 0.04) or having only sibling(s) above 2 years of age (*p* = 0.001) compared to no siblings, and childcare attendance (*p* = 0.005). Having sibling(s) only above 2 years of age versus no siblings was significantly associated with being qPCR-pos and Culture-pos compared to the qPCR-neg and Culture-neg (*p* = 0.001).

Multivariable logistic regression comparing clinical factors associated with being qPCR-pos and Culture-pos to being qPCR-pos only are shown in Table 3. There were no clinical factors that were statistically significant.

## 4. Discussion

In the post-PCV13 era, pneumococcal nasopharyngeal carriage has changed, and it is important to continuously monitor overall carriage in communities to understand the impact on clinical disease and the effectiveness of pneumococcal vaccination in controlling disease. This study of demographic and clinical features of children with and without *S. pneumoniae* carriage identified by culture or molecular methods, found no differences in features between children in whom *S. pneumoniae* carriage was found by qPCR only, compared with those who were positive by both culture and qPCR. As expected, there were differences in children who were positive by qPCR only, or by both qPCR and culture, compared with children who were negative by both methods, with positive children more likely to be 18 months of age, have siblings and attend daycare.

Since 2003, we have conducted thirteen point-prevalence studies measuring nasopharyngeal carriage in healthy children in Calgary using conventional culture methods [4,23]. These studies described not only a 92% decline in vaccine serotype carriage (from 77% to 6% of all serotypes identified) after the introduction of PCVs in the routine immunization schedule, but also a 34% decline in overall *S. pneumoniae* carriage (from 20% on average before 2010, to 13% in 2010–2012) [4]. Both culture and qPCR detection methods were used on samples from the two most recent surveys conducted in 2016 and 2018 [22]. *Streptococcus. pneumoniae* carriage was detected much more commonly with qPCR compared with culture methods (46% and 49% by qPCR, compared with 10% and 9% by culture for 2016 and 2018 surveys, respectively). Additionally, PCV13 serotypes were detected in larger proportions of *S. pneumoniae* positive samples with qPCR serotyping compared with Quellung serotyping (29% and 22% by qPCR, compared with 8% and 6% by Quellung for 2016 and 2018 surveys, respectively). These findings suggest that vaccine serotypes are still commonly present in healthy children despite universal PCV13 immunization in Alberta [22].

Thus, children are likely still an important reservoir for serotypes that were considered to be mostly eliminated in children using conventional identification methods. For example, there was a recent outbreak of serotype 4 *S. pneumoniae* in the homeless population of Calgary, despite serotype 4 not being seen in culture and Quellung based on carriage analysis in children [27].

The results of the current study are an extension of the microbiological findings from the 2016 survey [22], describing demographic and clinical factors in children which are associated with asymptomatic nasopharyngeal carriage of *S. pneumoniae*. No demographic or clinical factors were identified that significantly differed between the qPCR-pos and Culture-pos group compared to the qPCR-pos only group. This implies that there are no fundamental differences in the children that have *S. pneumoniae* nasopharyngeal carriage detectable only by qPCR methods compared with conventional culture methods. Therefore, these results can bridge the different methods for nasopharyngeal carriage testing in the pediatric population and justify comparing studies using culture or molecular detection methods. Future carriage studies using only molecular methods to both identify and serotype *S. pneumoniae* may be preferable as a more cost-effective option than using culture methods, that could require smaller sample sizes to accurately determine the prevalence of *S. pneumoniae* and serotypes, given the higher detection rate.

This study found that pneumococcal carriage was associated with age 18 months, having siblings, and childcare attendance. Having siblings has been consistently reported as a factor associated with increased pneumococcal carriage [4,28,29,30,31]. Daycare attendance has also previously been reported to increase pneumococcal carriage [4,28], as well as rural residence [17,31,32]; however, our study did not consider rural versus urban residence. Recent antibiotic use has a negative [17,29,32] or non-significant association [33], but the variation may be explained by the differing time periods used to define recent usage. We did not find any association between carriage and recent antibiotic usage amongst our study population, which can in part be attributed to the small number of children reportedly on antibiotics in each group. In an earlier study during the early post-PCV13 era, we did find reduced odds of carriage associated with recent antibiotic use [4]. That same study also found that two or more doses of PCV7 or PCV13 and older age were associated with reduced carriage *S. pneumoniae* carriage, a finding no longer seen during the era of routine three-dose PCV13 use only (compared with the prior use of routine four-dose PCV7 schedules) [4].

Limitations of our study include utilizing a convenience sample of children within restricted age categories under the age of five years old presenting to public health centers in Calgary for routine immunization, and thus our findings are particularly generalizable to a highly vaccinated population. However, herd immunity through indirect vaccine effects has been shown to have a protective effect in the unvaccinated population, and therefore, our results may still be generalizable to less highly vaccinated populations due to reduction in transmission to unvaccinated children from reduced carriage prevalence [23]. Additionally, we may be underestimating the true proportions of children for each clinical factor due to missing information in some questionnaires, since missing responses were not included in our analysis.

## 5. Conclusions

In summary, both culture and molecular methods are suitable to detect asymptomatic *S. pneumoniae* carriage in healthy children, and there are no fundamental differences in demographic or clinical features between children in whom *S. pneumoniae* is identified with either method. Since molecular detection is far more sensitive than culture, future carriage studies using only molecular methods to both identify and serotype *S. pneumoniae* may be preferable as a more cost-effective option, requiring smaller sample sizes to accurately determine the prevalence of *S. pneumoniae* and serotypes, given the higher detection rate.

## Figures and Tables

**Table 1 microorganisms-10-02074-t001:** Demographic and Clinical Factors of Study Participants From the 2016 Survey.

Factor	Total Survey n = 679 (%)	qPCR-pos and Culture-pos	qPCR-pos Only Any SP Carriage (PCV13 + NVT + NT) n = 242 (%)	qPCR-neg and Culture-neg
		Any SP Carriage (PCV13 + NVT + NT) n = 71 (%)		n = 322 (%)
**PCV13 doses ***				
0–1 dose	7 (1.0)	1 (1.4)	2 (0.8)	3 (0.9)
2+ doses	672 (99.0)	70 (98.6)	240 (99.2)	318 (98.8)
**Age**				
12 months	292 (43.0)	28 (39.4)	83 (34.3)	157 (48.8)
18 months	272 (40.0)	36 (50.7)	110 (45.4)	112 (34.8)
4.5 years	115 (16.9)	7 (9.9)	49 (20.2)	52 (16.1)
**Sex**				
Female	324 (47.7)	34 (47.9)	121 (50.0)	145 (45.0)
Male	355 (52.3)	37 (52.1)	121 (50.0)	176 (54.7)
**Siblings**				
0 sibling	256 (37.7)	19 (26.8)	75 (31.0)	142 (44.1)
At least 1 sibling <2 years of age	75 (11.0)	7 (9.9)	30 (12.4)	33 (10.2)
Sibling(s) only 2+ years of age	347 (51.5)	45 (63.4)	137 (56.6)	145 (45.0)
**Childcare attendance**				
Yes	186 (27.4)	22 (31.0)	80 (33.1)	72 (22.4)
No	492 (72.5)	49 (69.0)	162 (66.9)	248 (77.0)
**Underlying health conditions**				
Yes	47 (6.9)	5 (7.0)	23 (9.5)	17 (5.3)
No	632 (93.1)	66 (93.0)	219 (90.5)	304 (94.4)
**Otitis media in last year**				
Yes	110 (16.2)	13 (18.3)	43 (17.8)	45 (14.0)
No	565 (83.2)	57 (80.3)	198 (81.8)	274 (85.0)
**Hospitalized overnight in the last 6 months**				
Yes	10 (1.5)	2 (2.8)	4 (1.7)	3 (0.9)
No	669 (98.5)	69 (97.2)	238 (98.3)	318 (98.8)
**On antibiotics at time of survey**				
Yes	10 (1.5)	0 (0.0)	4 (1.7)	5 (1.6)
No	668 (98.4)	71 (100.0)	237 (97.9)	316 (98.1)
**On antibiotics in previous 2 months before survey**				
Yes	73 (10.8)	11 (15.5)	18 (7.4)	38 (11.8)
No	601 (88.5)	59 (83.1)	221 (91.3)	282 (87.6)

Results in columns may not add to 100% because of rounding error and missing responses. * Our study population selected for healthy children presenting for routine immunization, accounting for the large number of vaccinated children.

**Table 2 microorganisms-10-02074-t002:** Multinomial Logistic Regression of Clinical Factors Associated With *S. pneumoniae* Carriage Detected by qPCR and Culture Methods or qPCR Only Compared to Culture and qPCR negative.

Factor	qPCR-pos Only (n = 242) vs. qPCR-neg and Culture-neg (n = 322)			qPCR-pos and Culture-pos (n = 71) vs. qPCR-neg and Culture-neg (n = 322)		
	RRR ^†^	95% CI *	*p*-Value	RRR	95% CI	*p*-Value
**PCV13 doses**						
2+ doses vs. 0–1 dose	1.4	0.2–8.7	0.7	0.7	0.06–6.9	0.7
**Age**						
18 months vs. 12 months	1.7	1.1–2.5	0.009 ^§^	1.7	0.9–2.9	0.08
4.5 years vs. 12 months	1.4	0.8–2.3	0.2	0.6	0.2–1.4	0.2
**Siblings**						
At least 1 sibling <2 years of age vs. 0 siblings	1.9	1.0–3.6	0.04 ^§^	2.3	0.8–6.4	0.1
Sibling(s) only 2+ years of age vs. 0 siblings	2.0	1.3–2.9	0.001 ^§^	2.8	1.5–5.2	0.001 ^§^
**Childcare attendance**						
Yes vs. No	1.8	1.2–2.7	0.005 ^§^	1.7	0.9–3.1	0.09
**Underlying health conditions**						
Yes vs. No	1.8	0.9–3.6	0.09	1.3	0.4–3.7	0.6

^†^ RRR = Relative risk ratio. * CI = Confidence Interval. ^§^ Statistically significant.

**Table 3 microorganisms-10-02074-t003:** Multivariable Logistic Regression Comparing Clinical Factors Associated With Being qPCR-pos and Culture-pos versus qPCR-pos Only.

Factor	Odds Ratio	95% CI	*p*-Value
**Serotype carriage**			
PCV13 or related vs. NVT	0.6	0.3–1.2	0.2
**PCV13 doses**			
2+ doses vs. 0–1 dose	0.4	0.03–4.4	0.4
**Age**			
18 months vs. 12 months	1.0	0.6–1.8	0.9
4.5 years vs. 12 months	0.4	0.2–1.1	0.06
**Siblings**			
At least 1 sibling <2 years of age vs. 0 siblings	1.3	0.5–3.7	0.6
Sibling(s) only 2+ years of age vs. 0 siblings	1.4	0.8–2.7	0.3
**Childcare attendance**			
Yes vs. No	1.0	0.5–1.8	0.9

## Data Availability

The dataset for this study will be available in the University of Calgary Data Repository (PRISM Dataverse) and accessible according to the Canadian Tri-Agency Research Data Management (RDM) Policy.

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
