# Peer review of "Trends in Asymptomatic Nasopharyngeal Streptococcus pneumoniae Carriage with qPCR and Culture Analysis"

_microorganisms, 2022, doi:10.3390/microorganisms10102074_

Round 1

Reviewer 1 Report

Introduction section: The second paragraph repeats "in Alberta" twice within the same sentence. Restructure the sentence.

Results section: The total number of cases considered in the text indicates 682 while in table 1 it indicates that there are 679. I have not found an explanation for this difference in the text. Modify the frequencies and percentages so that the information in the results text and Table 1 match. Modify the statistics according to the corrected information.

Discussion section: It would be good to comment on the role of the nasopharyngeal microbiota in the NS carrier state.

Reviewer 2 Report

Review microorganisms 1969718

"Trends in Asymptomatic Nasopharyngeal Streptococcus pneumoniae Carriage with qPCR and Culture Analysis"

Streptococcus pneumoniae, the causal agent of human pneumonia, has an asymptomatic nasopharyngeal carriage in children. It is important to closely monitor and characterize (detection and serotyping) this reservoir and the consequence of the vaccination on it. Detection and serotyping of S. pneumoniae could be done by culture and with specific antisera (the quellum reaction) or using molecular methods such as qPCR. Because the qPCR detected higher level of carriage compared to the culture method, it is important to compare the clinical and demographic features of children who have S. pneumoniae carriage obtained by both methods. This is the purpose of this paper.

The paper is very well written. The authors used the samples collected in healthy children in Calgary in 2016. Such samples have been already studied, for instance in reference 22, the samples from 2016 and 2018 were analyzed by culture and qPCR and the authors compared both methods for the detection of S. pneumoniae and serotyping.

I have only few comments:

1- Materials and methods 2.2: It is not clear for me if the authors used the data from reference 22 or if they did again culture/Quellum reaction and qPCR from the swabs collected in 2016? This have to be indicated in the materials and methods.

2- Why the 2018 data were not used to strengthen the study?

3- Results presented in table 1 are different from those of reference 22: total survey 679 (table 1) vs 682 (reference 22), qPCR pos only 242 vs 243 and qPCR neg & culture neg 322 vs 323. Where these differences come from?

4- Table 2: Please check the first line of table 2 "RRR RRR+ RRR        RRR RRR Pvalue" Is that right?
